# Antioxidants and Collagen-Crosslinking: Benefit on Bond Strength and Clinical Applicability

**DOI:** 10.3390/ma13235483

**Published:** 2020-12-01

**Authors:** Franziska Beck, Nicoleta Ilie

**Affiliations:** Department of Conservative Dentistry and Periodontology, University Hospital, LMU, Goethestr. 70, D-80336 Munich, Germany; zahnaerztinfb@gmail.com

**Keywords:** antioxidants, adhesion to dentin, bond strength, Epigallocatechingallate, Chlorhexidindigluconate, Proanthocyanidin, Hesperidin

## Abstract

Antioxidants are known for their potential of strengthening the collagen network when applied to dentin. They establish new intra-/intermolecular bonds in the collagen, rendering it less perceptive to enzymatic hydrolysis. The study evaluated the benefit on shear bond strength (SBS) of a resin–composite to dentin when antioxidants with different biomolecular mechanisms or a known inhibitor of enzymatic activity are introduced to the bonding process in a clinically inspired protocol. Specimens (900) were prepared consistent with the requirements for a macro SBS-test. Four agents (Epigallocatechingallate (EGCG), Chlorhexidindigluconate (CHX), Proanthocyanidin (PA), and Hesperidin (HPN)) were applied on dentin, either incorporated in the primer of a two-step self-etch adhesive or as an aqueous solution before applying the adhesive. Bonding protocol executed according to the manufacturer’s information served as control. Groups (n = 20) were tested after one week, one month, three months, six months, or one year immersion times (37 °C, distilled water). After six-month immersion, superior SBS were identified in PA compared to all other agents (*p* < 0.01) and a higher reliability in both primer and solution application when compared to control. After one year, both PA incorporated test groups demonstrated the most reliable outcome. SBS can benefit from the application of antioxidants. The use of PA in clinics might help extending the lifespan of resin-based restorations.

## 1. Introduction

In the past, different authors presented evidence that the risk of replacement in posterior teeth is still higher for resin-based composite restorations than for amalgam fillings [1,2,3]. A loss in quality, especially in marginal adaption, can be noted as early as in the first three years after placement of a resin–composite filling [4]. The dentin–resin hybrid layer is considered the weak link in resin-based restorations [5], as dentin presents a very complex structure and morphology being streaked with microscopic tubules (~1–2 µm in diameter) dividing it into intertubular—rich in organic matter—and hypermineralized peritubular dentin [6,7]. Collagen type I represents the majority of the dentin organic material with around 90 wt.% [8]. While apatite crystallites are thought to provide strength, the collagen matrix provides toughness [9]. The collagen molecules exhibit a specific primary structure with repetition of the characteristic (X -Y- Gly)_x_ amino acid sequence, X and Y being mostly substituted by proline and 4-hydroxyproline residues [10,11,12]. Pashley [13] described intact dentin as a dynamic substrate as related to its differing distribution of minerals, organic network, water content, and permeability dependent on dentin depth, thus creating a unique bonding substrate dependent on the given distance to the pulp [14]. Age-related or pathologic changes in dentin structure and morphology as induced by sclerosis or caries pose additional challenges for dentin bonding [15,16]. Therefore, adhesive bonding to dentin is still considered technique sensitive [17], especially in clinical daily life, where, due to limited space in the oral cavity, possible contamination of the prepared cavity by saliva, blood or ubiquitous humidity is common [18]. Even under optimal conditions and with perfect control of the operating field, insufficient infiltration, incomplete solvent evaporation, and inadequate polymerization of the adhesive render the resin–dentin interface prone to degradation [5,19,20,21,22] and thus limit the longevity of resin–composite fillings [5]. The deterioration of the hybrid layer then proceeds via various mechanisms.

Water sorption results in plasticization of the polymer network [23], in hydrolysis of the resin components by broken ester linkages [24], and in disorganization of collagen fibrils [25]. In addition, exposed collagen fibrils are hydrolytically challenged by endogenous proteolytic enzymes, more precisely matrix metalloproteinases (MMPs) and cysteine cathepsins (CCs). Both protease families have been documented in both intact [26,27] and carious dentin [28] and show synergistic behavior: whereas MMPs operate best at neutral pH, CCs exhibit their highest activity at slightly acidic conditions, although for both proteinase families zymogen activation can be triggered by an acidic environment [29,30]. The activity of those host derived enzymes has been linked to not only the breakdown of the collagen network in carious lesions [28], but also beneath dental restorations [27] associated with acidic pretreatment during adhesive conditioning [31,32]. Consequently, this results in disintegration of the hybrid layer, which might further lead to restoration failure [27]. Therefore, to prolong the lifespan of adhesive restorations, one of the main focuses of dental research in recent years lay on how to eliminate proteolytic hydrolysis [33]. Primarily, two approaches have been pursued: (1) inhibition of host-derived enzymes, via chelation or blockage of the catalytic or allosteric center; and (2) cross-linkage of the dentin collagen network as inter- and intramolecular reinforcement renders the collagen matrix less assailable to enzymatic attacks.

Chlorhexidindigluconate (CHX) is the best researched MMP inhibitor in dentistry, as early studies showed beneficial effects in both periodontal and restorative applications [34,35]. CHX shows inhibiting effects on MMP-2, -8, and -9 and CCs [34,36], based on an unspecific chelating mechanism and interaction with sulfhydryl groups [34]. The usage of CHX against proteolytic degradation is especially interesting as it is already commonly applied in clinics due to its antibacterial and plaque reducing properties [37]. Similar to CHX, hesperidin (HPN) is also considered an MMP-/CC-inhibitor [38]. HPN is a glycosidic flavanone known for its antioxidant [39], anticarcinogenic [40], and antimetastatic [41] properties, which appear to be closely linked to its anti-MMP activity. In dental applications, HPN exhibited beneficial effects in reducing proteolytic degradation, preservation of the hybrid layer, and improvement of micromechanical properties of dentin [42,43,44]. In contrast, proanthocyanidin (PA) and epigallocatechin-3-gallate (EGCG) belong to the chemical group of polyphenolic flavonoids and are classified as crosslinking agents. Due to their structural analogy, the interaction with proteins, particularly collagen fibrils, is thought to be quite similar [45]. The abovementioned crosslinking agents share the potential of enhancing mechanical and physical properties of collagen matrices [46,47,48], inactivating MMPs, CCs, and their zymogenes [49,50,51], reducing proteolytic degradation. and maintaining the integrity of the hybrid layer, thus improving immediate and preserving long-term resin–dentin bond strength [42,46,51,52,53].

The objective of our study was to test the chosen agents on their effectivity in improving resin–dentin bond strength in a clinically relevant setting. The focus was on comparing differing operating principles and evaluating their efficiency. To render the bonding protocol time-efficient and simple, we decided for a two-step self-etch adhesive (Clearfil SE Bond 2, Kuraray Noritake, Japan) and integrated the bioactive agents either by administering them as an aqueous solution before the adhesive procedure or by incorporation in the primer of the adhesive system. The effective concentration was chosen after extensive literature review according to studies that already indicated benefit and efficiency for the abovementioned agents on bond strength and proteolytic hydrolysis [44,53,54,55]. Clearfil SE Bond 2 was chosen as adhesive system, as its mild-self etch formulation containing 10-MDP as functional monomer guarantees desirable dentin bond duration and was therefore recommended for dentin bonding by Van Meerbeek et al. [56,57,58] The innovation of this study is the direct comparison of the antioxidant agents with each other and CHX. Further, whether an application in an additional step might positively influence the substances effect on bond strength, improving their efficiency compared to direct incorporation into the primer formulation, was analyzed. As far as we are informed, a direct comparison of both primer incorporation and pre-adhesive application of these agents in a clinically applicable bonding protocol has not yet been conducted. In addition, the long observation period without accelerated aging is innovative in that context.

Our null hypotheses are stated as follows: (1) The integration of bioactive agents in the bonding protocol has no effects on the immediate or one-year bond strength. (2) There is no difference in bonding efficiency between the different application modes or among the bioactive agents.

## 2. Materials and Methods

### 2.1. Dentin Substrate Preparation

In total, 163 sound human third molars were collected from specialized oral surgery clinics and stored in sodium azide solution (0.2%) at 20 °C and used within four weeks after the beginning of the experiments. The roots were separated from the crown through a cut slightly below the cemento-enamel junction with a low-speed diamond saw (IsoMet, Buehler; Lake Bluff, IL, USA). Then, the crown was cut slightly above the tooth equator into two halves, a coronal and a cervical one, exposing the same dentin surface. Furthermore, each half was cut in four parts representing the mesiobuccal, mesiolingual, distobuccal, and distolingual corners and resulting at optimum in eight parts per tooth in total (tooth parts that did not present the needed dentin surface for standardized bonding due to differing tooth size were discarded). Each tooth part was embedded in methacrylate resin (Technovit 4004, Kulzer, Hanau, Germany, Liquid LOT R010050, Powder LOT R010031) in a stainless-steel cylinder (diameter = 16 mm) and stored in distilled water at 37 °C for maximum 72 h. The specimens were then randomly allocated to their respective test groups. To obtain a standardized smear layer and a flat dentin surface, the samples were wet ground with 600 grit SiC-paper (Leco, St. Joseph, MI, USA) on a grinding system (Exakt 400 cs, Norderstedt, Germany) for 30 s. A one-sided adhesive paper was applied on the samples standardizing the bonding area to a circle round of dentin with a diameter of 3.16 mm (Figure 1).

### 2.2. Bonding Procedure

The abovementioned agents were integrated in the bonding process by being added either to the primer of a commonly used self-etch adhesive (Clearfil SE Bond 2, Kuraray Noritake, Chiyoda Japan, LOT 000031) (Table 1) or before the primer applied as an aqueous solution.

HPN (purity level (PL) ~ 90%, LOT 116241478) was obtained from Carl Roth (Karlsruhe, Germany), while PA (no PL available, LOT F0I066), EGCG (PL > 95%, LOT # SLBL3516 V) and CHX (aqueous solution, concentration = 20%, LOT #LRAA9090) were purchased from Sigma Aldrich (Darmstadt, Germany). For the testing solutions, the agents were dissolved in their respective concentrations (EGCG = 100 µM, CHX = 2%, PA = 3.75%, and HPN = 5%) either in the primer of Clearfil SE Bond 2 or in distilled water. Furthermore, the testing solutions were kept in UV/Light-protective bottles, stored at 4 °C and used 24 h after being mixed. No additive was added to the bonding process for the control group. Each of the nine test groups (every agent solved each in primer and distilled water plus control group) was tested after 1 week, 1 month, 3 months, 6 months, and 1 year of immersion duration (distilled water, 37 °C). That sums up to a total of 45 test groups of 20 specimens (Figure 2).

The shear bond strength (SBS) was evaluated in a shear test. For the primer test group (EGCG-/CHX-/PA-/HPNp), the test primer incorporated with the respective additive in the respective concentration was applied on the exposed area (diameter = 3.16 mm; area = 7.84 mm²) and worked in with a microbrush for 20 s. Afterwards, the test primer was air-dried for 5 s and bonding agent was applied, dispersed, and polymerized for 10 s with an LED-curing-Unit (Bluephase LED, Ivoclar Vivadent, Ellwangen, Germany, 792 mW/cm²) as advised by the manufacturer’s information. Before polymerization, a mold with a height of 4 mm was placed on each specimen, exposing only the bonding area (diameter = 3.16 mm) and standardizing the curing position. For the solution test groups (EGCG/CHX/PA/HPN), the test solution containing the respective agent was worked in for 20 s and gently dried with air for 5 s, and then the unaltered primer and bonding agent were applied and polymerized as described above. A low-shrinkage ORMOCER (organically modified ceramics)--bulk-fill resin–composite (Admira Fusion xtra, VOCO, Cuxhaven, Germany, LOT 1537600) was used as a restoration material, applied into the mold and condensed to a height of 3 mm (Figure 1). The restorative was light-cured as described in the manufacturer’s information (792 mW/ cm², 40 s). Afterwards, the adhesive paper was removed, and the specimens were stored in distilled water at 37 °C in a thermal oven (Jouan EU3, INNOVENS Ovens, ThermoFisher Scientific; Waltham, MA, USA) until their respective testing after 1 week, 1 month, 3 months, 6 months, or 1 year. The distilled water was changed regularly every two weeks without disturbance of the specimens.

The testing was carried out with a Universal testing Machine (MCE2000ST, Quicktest Prüfpartner GmbH, Langenfeld, Germany) with a knife-edge chisel device at a crosshead speed of 0.5 mm/min until fracture (Figure 3).

To ensure thorough polymerization, the irradiance (792 mW/cm²) of the LED-curing-Unit (Bluephase LED, IvoclarVivadent, Ellwangen, Germany) was measured before the start of the experiments with a spectrophotometer system MARC (Managing Accurate Resin Curing, Bluelight Analytics Inc., Halifax, Nova Scotia, CAN) at a height of 4 mm, imitating the distance to the bonding area created by the usage of a mold.

### 2.3. Fractography

The breaking mechanism was analyzed with a magnifying glass with a 10× magnification and defined as follows: an adhesive failure constitutes a fracture pattern that is located exactly between the tooth and resin composite; a mixed fracture shows partly an adhesive failure but proceeds either through one or both substrates (dentin and resin–composite); and a cohesive break is indicated when a fracture line runs only in one or both substrates and not along the bonding area.

### 2.4. Statistical Analysis

The data were analyzed with a multivariate analysis of variance (general linear model, partial eta-squared statistics) which assessed the influence of the parameter agent (EGCG, CHX, PA, HPN, and control), application (solution, primer, and control), and the parameter immersion duration (1 week, 1 month, 3 months, 6 months, and 1 year) on the SBS value. Furthermore, the data were statistically evaluated with a one-way ANOVA (analysis of variance) and the Tukey HSD (honestly significant difference) post-hoc test. All data were checked for normality by the Shapiro–Wilk-Test and all statistical tests were executed at a confidence level of 95.0% (IBM SPSS, Version 24.0; Armonk, NY, USA). The study protocol and the sample size were chosen and refined after preliminary studies. The power of the sample size was calculated and rechecked after one-week immersion on the basis of mean and standard deviation of the bond strength values for group CHXp and HPNp, which resulted in a power of 85.63% for a sample size of 20.

The data were further analyzed using Weibull analysis. The Weibull distribution is a common model used to assess the cumulative probability of default P for brittle materials at applied stress:(1)$$P_{f}\left(\sigma_{c}\right)\;=\;1-exp{\left[-\left(\frac{\sigma_{c}}{\sigma_{0}}\right)\right]}^{m}$$
where $\sigma_{c}$ is the measured strength at failure, $\sigma_{0}$ is the charateristic strength which is defined as the strength at which $P_{f}$ equals 0.632, and m is the Weibull modulus. The double logarithm of the before mentioned equation results in the following expression:(2)lnln11−P = mlnσc− mlnσ0

The Weibull modulus m is the upward gradient of the straight-line graph, resulting by plotting lnln11−P  against  lnσ.

A confidence interval (*CI*) for a confidence level of 95% for the Weibull modulus m was computed by calculating the standard error (*SE*):(3)$$SE\;=\;m\;\frac{\sqrt{1-R^{2}}}{\sqrt{R^{2}\;\left(N-2\right)}}$$
where *m* is the Weibull modulus, $R^{2}$ is the coefficient of determination, and *N* is the number of tested specimens:(4)CI = SE · θ ±m

When the confidence level is defined as 95%, θ equals 1.96 in normally distributed data.

### 2.5. Ethical Approval

This study does not include any experiments involving human participants or animals performed by any of the authors. In relation to ethical guidelines, the human teeth used represent residual biological material. For this kind of study, there is no consultation obligation by the institutional ethics committee, as stated in the §24, 2 medical products law. The study was approved under the project number 19- 535 KB.

## 3. Results

Using a multifactorial, univariate analysis of variance (ANOVA, *p* ≤ 0.05) with partial eta-squared statistics, the influence of the parameters “agent”, “application mode”, and “immersion time” on bond strength was assessed. The parameter “agent” showed the strongest impact on SBS (*p* < 0.001; η^2^_p_ = 0.067); “immersion time” (*p* < 0.001; η^2^_p_ = 0.042) also significantly influenced SBS, but to a lesser extent. Contrarily, there was no significant effect for “application mode” (*p* = 0.228) on bond strength. Interestingly, the combination “agent and application mode” (*p* < 0.001; η^2^_p_ = 0.034) displayed a significant effect on bond strength, suggesting a dependence between those parameters. The binary combinations “agent and immersion time” and “application and immersion time” as well as the ternary combination “agent and application mode and immersion time” showed a significant (*p* < 0.001; *p* < 0.001; *p* = 0.023) but small influence (η^2^_p_ = 0.040; η^2^_p_ = 0.024; η^2^_p_ = 0.027) on SBS.

For immediate bond strength, after one week of immersion, the ANOVA did not display statistically significant differences (*p* = 0.199) among the test groups. In post-hoc analysis, there was no significant difference between the control group and any test group (*p* > 0.05). However, Weibull analysis showed more reliable bond strength values for both HPNp (m = 4.35 ± 0.25) and control (4.67 ± 0.39) in comparison to all remaining test groups. Except for immediate bond strength (*p* = 0.199), the conducted ANOVAs showed a highly significant impact of the test groups on SBS (*p* < 0.001) for all further immersion intervals.

After one month of immersion, PAp, HPNp, and control all demonstrated significantly higher SBS values when compared to EGCGp (PAp: *p* = 0.013; HPNp: *p* = 0.028; control: *p* = 0.003) and EGCGs (PAp: *p* = 0.001; HPNp: *p* = 0.003; control < 0.001). CHXs however only displayed statistically superior bond strength values when compared to EGCGs (*p* = 0.039). Even though PAp, CHXs and HPNp were not statistically different from control after one month of immersion time, Weibull analysis indicated a higher reliability for the SBS values of PAp (m = 5.03 ± 0.62) and CHXs (m = 4.06 ± 0.47) in comparison to control (m = 3.24 ± 0.36) and HPNp (m = 2.77 ± 0.29).

After three months of immersion time, PAp provided statistically superior data (*p* < 0.05) compared to all test groups except for CHXs (*p* = 0.258), PAs (p = 0.136), and HPNp (*p* = 0.517). However, PAp’s Weibull modulus (m = 9.41 ± 0.99) exhibited a higher reliability than any other test group including CHXs (5.66 ± 0.44), PAs (3.23 ± 0.17), and HPNp (m = 2.46 ± 0.39). Thus, PAp was significantly different from (*p* = 0.013) and further conducted more reliable results than control (m = 2.27 ± 0.30). After six months of immersion, post-hoc analysis revealed significant higher SBS values for both PA test groups compared to EGCGs (PAp: *p* = 0.019; PAs *p* = 0.034), CHXp (PAp: *p* = 0.002; PAs *p* = 0.004), and HPNp (PAp: *p* = 0.006; PAs: *p* = 0.011). However, post-hoc analysis did not provide a statistically significant difference in SBS among PAp, PAs, and control, although PAp presented a significantly higher Weibull modulus (m = 6.12 ± 0.86) than both PAs (m = 3.95 ± 0.21) and control (m = 2.79 ± 0.28).

After one year, control only exhibited superior bond strength when compared to EGCGp (*p* = 0.004). EGCGp itself displayed a poor medium-term performance with a mean of 8.42 MPa and was hence significantly inferior in comparison to all test groups (*p* < 0.05) with the exception of CHXp (*p* = 0.928). PAp and PAs provided higher SBS values than EGCGp (PAp: *p* < 0.001; PAs: *p* < 0.001) and CHXp (PAp: *p* = 0.041; PAs: *p* = 0.006), and both demonstrated more reliable results (PAp: m = 5.72 ± 0.60; PAs: 6.12 ± 0.86) compared to any other test group.

The influence of immersion duration on the SBS of the test groups was analyzed by one-way ANOVAs and Tukey HSD. In post-hoc analysis, apart from EGCGp, which showed a significant loss of bond strength in between one-week and one-year SBS (*p* = 0.011), all test groups, including control (*p* = 0.996), maintained their bond strength when one-week and one-year SBS values were compared. Furthermore, while the solution test groups (EGCGs/ CHXs/ PA/ HPNs) exhibited consistent values throughout all immersion intervals, all primer-solved groups and control attained statistically varying results in post-hoc analysis (Table 2).

The Weibull modulus m describes the reliability of the tested SBS for each test group. It is depicted as the upward gradient of the straight-line graph, resulting by plotting lnln11−P  against  lnσ. Therefore, the higher is m, the less scattering there is of individual measurements and thus the more accurate do the measured data represent the SBS of the respective test group. Further, for a Weibull modulus m≈3.602, data distribution resembles normal distribution. The calculated Weibull moduli for many test groups approximate this value (Table 3), which confirms the result of the Shapiro–Wilk-test for normal distribution of the data.

Most test groups show a similar development for the characteristic strength $\sigma_{0}$ in relation to immersion times: At the beginning, $\sigma_{0}$ is rising until it hits its maximum at 1–3 months immersion time (dependent on the respective test group) followed by a decline down to one year of aging. While PAp, PAs, and control show a gradual decline thereafter, EGCGs, CHXp, CHXs, and HPNp display their minimum values at six months, and then again rise in characteristic strength up to one year of immersion. The lowest characteristic strength of all test groups is noted with EGCGp after one-year aging.

When comparing the graphical illustrations (Figure 4) between solution-based and primer-based test groups after one week and one year of immersion, an inverted development is revealed. While the primer test groups present relatively steep graphs in narrow alignment for one-week testing, after one year a distinctive flattening of the graphs’ slope and a broader scattering of the m-values may be noted. The solution-based test groups, by contrast, display an exactly opposite development, as the Weibull moduli seem to increase between one week and one year of immersion. This hints towards an advantage for a solution-based bonding protocol in long-term results.

As R² ranges on a high level between 0.76 and 0.99, presenting predominantly values > 0.9, the Weibull regression analysis proves to be a good model fit to predict the data development for the chosen experimental setup (Table 3).

Assessing the fracture pattern of all tested substrates (900) irrespective of the test groups, the predominant break occurred in the adhesive layer (54.7%; 492 substrates); 43.6% (392 substrates) were further classified as mixed, only 1.8% (16 substrates) as cohesive, and no premature failure was registered. As a general trend, it can be observed that substrates with high SBS values tend to be associated rather with a mixed or cohesive fracture pattern than to show adhesive failure. Detailed data concerning fracture analysis are presented in Table 4.

## 4. Discussion

MMPs and CCs form families of proteolytic enzymes known and specialized for the degradation of extracellular matrix (ECM) [59,60] and are vastly expressed by human odontoblasts and pulp tissue [26,61]. However, definitive occurrence and proteolytic activity in both carious and intact dentin was only determined for cysteine cathepsin B (CC-B) and cystein cathepsin K (CC-K) [28,62,63], while the most abundantly found metalloproteinases in dentin are MMP-2, -8, -9, and -20 [64,65,66]. Both enzyme families can be triggered in acidic environment; however, while CCs exhibit highest proteolytic activity at low pH, MMPs are known to be neutral proteinases [29,30]. As there is indication of reciprocal activation with CC and MMPs [67,68] and Scaffa et al. further demonstrated close proximity between CC-B and MMP-2 on the collagen fibril [63], a close interaction and cooperation of both proteinase families in the proteolytic degradation of the ECM has to be assumed. Therefore, collagenolysis in dentin in carious lesions and beneath dental restorations is probably the result of the close interaction between CCs and MMPs. The proteinases are activated by low pH environment induced either by bacterial acids of cariogenic, dental plaque [28,29,62], or cavity conditioning, as demonstrated for both etch-and-rinse and self-etch systematics [31,32].

While for CHX, PA, and EGCG previous research demonstrated anti-proteolytic activity focused on the interaction with dentin MMPs and CCs, HPN exhibits general anti-enzymatic activity against intrinsic collagenolytic activity in dentin [36,38,69,70]. HPN is a bioflavonoid from the subgroup of glycosidic flavanones, abundantly available in citrus fruits. Various studies confirmed beneficial antioxidant [39], anticarcinogenic [40], and antimetastatic [41] effects, which indicate a close correlation with HPN’s anti-MMP activity. In dental applications, HPN exhibited beneficial effects in reducing proteolytic degradation, preservation of the hybrid layer, and improvement of micromechanical properties of dentin [42,43,44]. Islam et al. [44] showed improved immediate and long-term (tested after one year of immersion) bond strength when 2% HPN was incorporated into the primer of a commercially available self-etch adhesive. The incorporation of 5% HPN did not improve µTBS immediately or after one year, but, in contrast to 2% HPN, it preserved bond strength and fibril morphology after one-year immersion. Based on that research, we decided on the test concentration of 5% for our HPN groups, as the preservation of collagen fibrils is crucial for longer-term adhesive success [27]. Our data present a preferable outcome for HPNp up to three months of immersion: After six months, however, HPNp exhibited a significant decline (*p* = 0.001) in bond strength of 38.6% to 10.10 MPa in comparison to the three-month SBS. Restabilization of bond strength (13.12 ± 4.92) MPa was shown after one year of immersion. However, similar to the results presented by Islam et al. [44], neither for HPNp nor HPNs a significant difference for neither immediate nor medium term (measured after one-year immersion) could be observed when compared to control. In general, HPNs presented more stable bond strength values throughout the test period (Table 1). Although HPN was initially categorized as crosslinking agent [71], recent research contradicts this theory as neither the triple-helical structure nor thermal stability of collagen was altered by the administration of HPN [72]. Therefore, the authors of [72] formulated the theory HPN might bind loosely to dentin collagen, blocking the cleavage sites of proteinases and hence prevent proteolytic degradation as previously demonstrated [42]. The loose configuration of the bond might, inter alia, be responsible for the decline in bond strength of HPNp after three months of immersion. However, since HPNs does not show the same loss in SBS and demonstrates stable results throughout the immersion period, it can be speculated that the performance of HPN on dentin might be influenced by the respective solvent and the solution’s pH. An explanation might be a possible precipitation of HPN at low pH environment, as observed by Gil-Izquierdo et al. [73]. Even though the exact molecular mechanism for collagen–HPN interactions is yet to be determined, the literature review indicates both hydrophobic and hydrophilic interactions attributed to HPNs amphiphilic structure, however the hydrophobic mechanism seems to be more pronounced [42,44,72,74,75].

HPN as well as PA and EGCG are counted as bioflavonoids as they share a benzene-pyran-phenolic acid as common structural component. However, PA and EGCG additionally contain polyphenolic moieties as structural characteristic and are therefore classified as tannins, a subfamily of bioflavonoids. Both PA and EGCG are considered collagen crosslinkers and prove their efficiency in dentin collagen biomodification in previous research: enhancing mechanical and physical properties of collagen matrices [47,48], reducing proteolytic degradation, and maintaining the integrity of the hybrid layer, thus improving immediate and preserving long-term resin–dentin bond strength [42,52,53]. EGCG, the predominant catechin in green tea [76], demonstrated very favorable results and a high efficiency when used in a time-efficient protocol at a low concentration. Khamverdi et al. incorporated EGCG in the primer of a common self-etch adhesive (Clearfil SE Bond, Kuraray Noritake, Chiyoda, Japan) without adverse effects on the degree of conversion and thereby preserved the dentin bond for the concentration of 100 µM [53]. PA is, besides glutarylaldehyd (GA), the most extensively studied crosslinking agent in dental applications. However, in contrast to GA, it demonstrates the valuable trait of low toxicity and good tolerability [48]. PA is a naturally derived agent, abundantly found in grape seeds. Proanthocyanidins are in fact a heterogenous group of tannins consisting of oligomeric or polymeric procyanidins, which are galloylated to various degrees [77]. For our study, we utilized purified grape seed proanthocyanidins ordered from Sigma Aldrich, Germany. Due to their structural analogy, PA and EGCG display a very similar molecular mechanism when binding to collagen: hydrogen bonds are the predominant driving force for protein–PA/EGCG complexes [77,78], being established between the protein amide carbonyl and phenolic hydroxyl. PRPs present a special affinity for tannins due to the electron distribution of the peptide bond that results from the substituted nitrogen adjacent to the carbonyl group, rendering PRPs particularly strong hydrogen bond acceptors [79]. Furthermore, PRPs, such as gelatin or collagen, exhibit a very accessible tertiary structure, e.g., helix-structure or a random coil [80,81], that further eases the bond formation due to the exposition of the carbonyl oxygen [79]. Hydrogen bonds are additionally stabilized by adjacent hydrophobic pi-stacking between the proline’s pyrrolidine and the phenolic rings rendering the crosslinks resistant against aqueous buffers [48,82]. In addition, the effect on proteolytic enzymes, namely MMPs and CCs, might be explained by the affinity of tannins (EGCG and PA) for proline residues: CC-K is known for its unique collagenolytic activity among mammalian proteinases, as it does not depend on destabilization of the collagen helix and is able to cleave crosslinked collagen Type I and II at multiple sites within the triple-helical region of the peptide [83]. As mentioned above, CC-K is one of the few CCs whose proteolytic activity has been definitively detected in both carious and intact dentin [28]. Interestingly, CC-K exhibits an exclusive specificity for a prolin residue at the P2 position of its substrate. Mutation of the S2-subsite of CC-K into an CC-L-like subsite impaired the collagenolytic activity of CC-K, as demonstrated by Lecaille et al. [84]. Thus, it is reasonable to assume that the preferential interaction of PA and EGCG with prolin residues of the collagen helix might block CC-K from binding to the peptide and therefore inhibit its collagenolytic activity. EGCG/ PA might further interact with the prolin-rich hinge region of MMPs, forming complexes that impede the flexibility and mobility of the enzyme [85,86].

Despite the structural analogy, and in contrast to previous reports [53], our results show unstable and inferior outcome for EGCG particularly when compared to PA. Even though there is no loss in SBS documented for either EGCGs or EGCGp for immediate bond strength, after one month both EGCGp and EGCGs display significantly lower SBS in comparison to control (EGCGp: *p* = 0.003; EGCGs: *p* < 0.001) and PAp (EGCGp: *p* = 0.013; EGCGs: *p* = 0.001). After one-year immersion, EGCGp exhibits significantly inferior SBS values compared to control (*p* = 0.004), PAp (*p* < 0.001) and PAs (*p* < 0.001), and additionally both EGCGp and EGCGs present less reliable results than control, Pap, and PAs, as confirmed by Weibull analysis (Figure 2). The malperformance of EGCG in comparison to PA might be explained by a molecular weight effect. Frazier et al. proved lower binding constants for tea catechins compared to oligomeric polyphenols such as grape seed PAs. As larger molecules offer more binding sites to proteins such as gelatin or BSA, the interaction is stronger, hence the complex formation occurs more readily than to monomeric polyphenols [77,87]. In addition, even though both concentrations were chosen after previously successful, time-efficient study setups, the total applied amount of substance of EGCG per specimen is of course much lower compared to that of PA due to the differing concentrations that were used (EGCG: 100 µM (wt./vol); PA: 3.75% (wt/vol)). This might explain the less efficient performance compared to PA; however, it does not unravel the malperformance in comparison to the control group and its general adverse effect on bond strength. PA contrarily shows very viable results throughout the test period, for both test groups PAs and PAp. Interestingly, however, while PAp performs exceedingly well up to three months, a slight decline in SBS is seen after six months of immersion. This might hint at a possible interaction between components of the primer and PA which seems to promote crosslink formation at first, but whose beneficial influence decreases after a certain time. Generally, Hagerman et al. showed best PA–protein interaction near the isoelectric point of the respective protein, as the electrostatic repulsion between the protein molecules is then minimized [79,88]. The isoelectric point of collagen is localized around pH 7.8 [89]. This might explain the more consistent results for PAs throughout immersion. However, both PAp and PAs are the only test groups that show benefit on one-year bond strength, since their Weibull modulus indicate more reliable outcome compared to any other test group, including control.

Since CHX is the most extensively studied inhibitor of proteolytic activity in dentistry, we introduced CHX in a 2% (vol/vol) concentration as negative control to our study [55]. However, no improvement of the SBS values for immediate bond strength or after one year could be deducted by the administration of CHX. Interestingly, however, CHXp exhibited varying outcome throughout the test intervals. After six months of immersion, CHXp provided significantly inferior SBS than control (*p* = 0.010) and after one year less reliability (m = 2.57 ± 0.28) compared to control (m = 4.00 ± 0.25), whereas CHXs showed steady and reliable results throughout the test period and even displayed significantly higher SBS (*p* = 0.022) compared to CHXp after one year of aging. This is in accordance with the results of Hiraishi et al. [90], who showed adverse effects for the incorporation of 2% CHX in the primer of a self-etch primer of a luting cement (ED primer, Panavia F 2.0, Kuraray Noritake, Japan). Since CHX, due to its single or double positive charge over a broad pH range (pKa values 2.2/10.3), forms strong electrostatic bonds with anionic functional groups, e.g., phosphate groups of hydroxyapatite [91], CHX might also interact with the anionic phosphate groups of 10-MDP, impairing the formation of ionic bonds between the functional monomer and calcium ions and therefore impeding the bonding performance of the adhesive [90,92]. As Clearfil SE Bond 2 also contains 10-MDP as functional acidic monomer, the theory might explain the malperformance of CHXp compared to CHXs.

As pointed out in the Results chapter, the fractographic analysis suggests a correlation between high bond strengths with mixed or cohesive fracture patterns: 74% of all mixed breaks progressed only through resin–composite, which could indicate that the restoration material used represented a weaker pathway facilitating crack propagation. However, the ratio of mixed fractures occurring in composite was very similar not only for overall test data excluding control (75.5%) but also showed similar distribution in PA test groups (74.7%). Consequently, the conclusion that fracture continues preferably through the resin composite due to an intrinsic cohesive toughening effect on dentin collagen through the application of the test agents is not endorsed.

The common point of critique about shear bond tests exhibiting a high percentage of cohesive failures [93] cannot be confirmed by this study because only 1.8% of all breaks were classified as cohesive. 

Although bond strength testing with ageing protocols correlate reasonably with medium-term retention rates [93], the obtained data should be interpreted with caution and in relation to the used methodology. Bond strength tests show an inhomogeneous distribution of stress at the adhesive interface and are thus highly influenced by experimental conditions [93,94,95].

All things considered, both null hypotheses have to be rejected: even though none of the test agents did improve immediate SBS after one week of immersion, the administration of PA definitively demonstrated advantage for bond strength and its preservation when applied in a time-efficient protocol. Additionally, both PA application modes seem to be viable choices for integrating PA into adhesive bonding. Generally, it might be advisable to administer bioactive agents in a separate step, since possible interaction with the formulation of the primer and thereby impairment of its bonding performance can be avoided. This is supported by our results, as the test groups administered as aqueous solution indicated more stable SBS throughout the test periods. Furthermore, solvent and pH of the solution can be better adjusted to the chemistry of the respective agent, enforcing or facilitating either the formation of crosslinks or the inhibition of proteolytic enzymes such as MMPs or CCs, thus rendering the interaction more effective.

The results obtained from this study might encourage the introduction of antioxidants, especially PA, to commercial products. In addition, adhesive cementation of indirect resin or ceramic-based restorations could benefit from their positive effect on the dentin network and in extension stability of the hybrid layer. In our opinion, more basic research on how the micro-molecular processes of collagen crosslinking and the interaction between proteolytic enzymes and inhibiting agents work in detail would benefit future research considerably.

## 5. Conclusions

Within the general limitations of a laboratory study, the results indicate possible advantages for the introduction of PA in adhesive dentistry as it may prolong the lifespan of direct resin-based restorations, thus reducing the replacement frequency further leading to cost reduction and preservation of dental hard tissue. Further, separate application of the discussed agents led to more reliable bond strength results after one year, although there was no improvement seen for immediate bond strength. The study also stresses the importance of considering the chemical structure of an additive before incorporating it in the bonding process in order to avoid negative interactions with the formulation of the adhesive.

## Figures and Tables

**Figure 1 materials-13-05483-f001:**
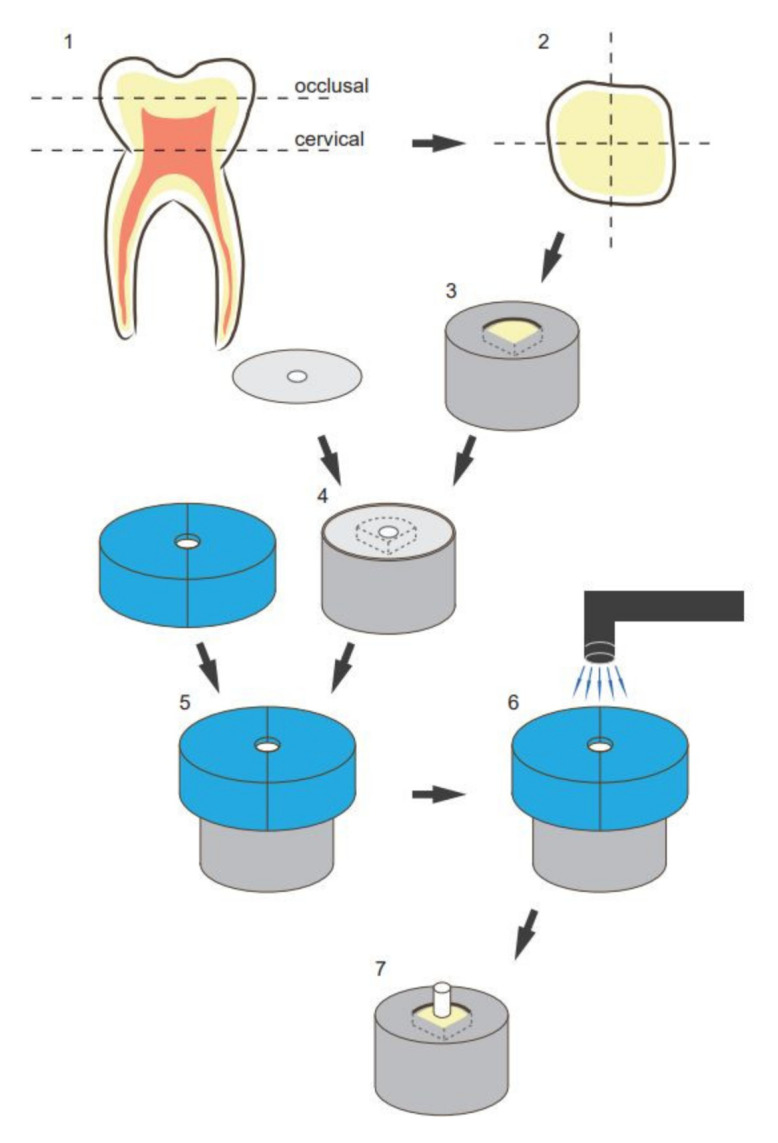
Illustration of sample preparation and bonding procedure: (**1**) schematic presentation of the cutting procedure; (**2**) schematic presentation of the cross-section of each tooth half; (**3**) embedding of the dentin substrate in methacrylic resin; (**4**) placement of the adhesive paper (with a centered circle-round hole; diameter of 3.16 mm) on the specimen to limit the bonding area; (**5**) placement of the vinyl polysiloxane split mold with a cylindric cavity (diameter of 3.16 mm and height of 4 mm); (**6**) schematic presentation of the resin–composite placement and light curing; and (**7**) restored final specimen.

**Figure 2 materials-13-05483-f002:**
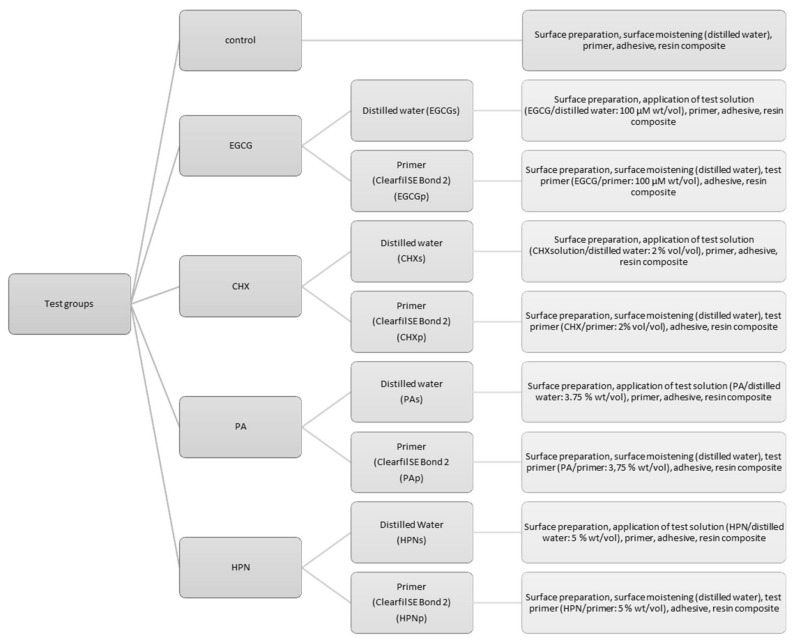
Description of the test groups; each tested after one week, one month, three months, six months, and one year.

**Figure 3 materials-13-05483-f003:**
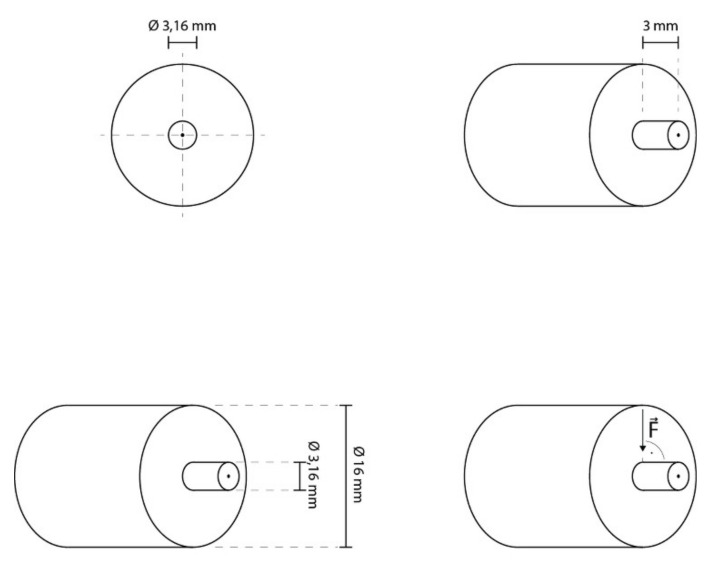
Graphics illustrating specimen dimensions and testing.

**Figure 4 materials-13-05483-f004:**
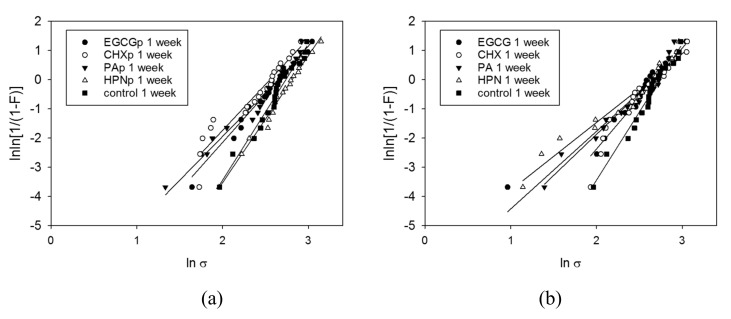
Example of Weibull graphs for the shortest (one week) (**a**,**b**) and longest (one year) (**c**,**d**) immersion durations and both antioxidant application protocols: (**a**) one week, primer; (**b**) one week, solution; (**c**) one year, primer; and (**d**) one year, solution.

**Table 1 materials-13-05483-t001:** Characterization and composition of materials.

Commercial Name, Manufacturer, LOT Number	Type of Material	Main Components	Instructions for Use
**Clearfil SE Bond 2,** **Kuraray Noritake,** **LOT 000031**	Two-step self-etch adhesive	Primer: 2-Hydroxyethylmethacrylat 10-Metharyloyloxydecyl-dihydrogenphosphate Hydrophilic aliphatic dimethacrylate dl-campherchinone accelerators water dyes	Apply primer to the entire cavity wall for 20 s and dry with mild air for more than 5 s until the PRIMER does not move;
		Adhesive: bisphenol A diglycidylmethacrylate 2-hydroxyethyl metharylate 10-methacryloyloxydecyl dihydrogen phosphate Hydrophobic aliphatic dimethacrylate Colloidal silica dl-campherquinone initiators accelerators	Apply bond to the entire cavity wall and make a uniform bond film using a gentle air flow;Light-cure bond with a dental curing unit for 10 s
**Admira Fusion xtra, VOCO, LOT 1537600**	Nanohybrid-ORMOCER bulk-fillresin-composite	Matrix:ORMOCERFillers:Based on silicon oxide (84 wt.%)	Apply in ≤ 4 mm increments;Light cure for 40 s

**Table 2 materials-13-05483-t002:** Mean and standard deviation of SBS as a function of immersion duration; *p*-values signify the ANOVA results of each column/row; different lowercase letters indicate significant differences in a column tested by Tukey’s post-hoc analysis.

Bond Strength (MPa)	EGCG 100 µM (wt/vol)		CHX 2% (vol/vol)		PA 3.75% (wt/vol)		HPN 5% (wt/vol)		Control (No Additive)	ANOVA
	Primer	Solution	Primer	Solution	Primer	Solution	Primer	Solution		
One week	12.68 (4.31) a	12.53 (4.27) a	11.41 (3.84) ab	13.18 (4.47) a	12.49 (4.06) a	12.66 (4.28) a	15.12 (3.92) a	12.10 (4.77) a	14.10 (3.40) ab	*p* = 0.199
One month	11.78 (3.41) ab	10.84 (4.28) a	13.80 (5.84) a	15.59 (3.94) a	17.03 (4.08) bc	13.53 (4.16) a	16.68 (5.72) a	15.35 (3.65) a	17.66 (5.89) a	*p* < 0.001
Three months	12.30 (5.02) a	13.29 (5.37) a	11.50 (3.25) ab	15.84 (3.23) a	19.23 (2.37) c	15.43 (5.03) a	16.42 (5.39) a	12.59 (4.27) a	14.29 (4.14) ab	*p* < 0.001
Six months	12.11 (3.52) a	10.56 (3.82) a	9.69 (3.81) b	12.18 (4.71) a	15.08 (3.16) ab	14.85 (4.15) a	10.10 (4.47) b	12.68 (4.19) a	14.49 (5.06) ab	*p* < 0.001
One year	8.42 (3.83) b	13.35 (5.06) a	10.41 (3.92) ab	14.63 (5.30) a	14.36 (2.95) ab	15.17 (2.98) a	13.12 (4.92) ab	14.41 (3.93) a	13.58 (3.66) b	*p* < 0.001
ANOVA	*p* = 0.009	*p* = 0.159	*p* = 0.027	*p* = 0.043	*p* < 0.001	*p* = 0.178	*p* < 0.001	*p* = 0.074	*p* = 0.046	

**Table 3 materials-13-05483-t003:** Weibull modulus m, confidence interval (95%), coefficient of determination R^2,^ and characteristic strength $\sigma_{0}$ calculated for each test group.

Weibull Modulus m, Confidence Interval (95%), R^2^ and σ_θ_ (MPa)	EGCG 100 µM (wt./vol)		CHX 2% (vol/vol)		PA 3.75% (wt./vol)		HPN 5% (wt/vol)		Control (No Additive)
	Primer	Solution	Primer	Solution	Primer	Solution	Primer	Solution	
One week	3.30 (0.24) R^2^ = 0.98 σ_0_ = 14.15	2.66 (0.37) R^2^ = 0.92 σ_0_ = 14.35	3.15 (0.44) R^2^ = 0.92 σ_0_ = 12.82	3.46 (0.46) R^2^ = 0.92 σ_0_ = 14.67	3.00 (0.28) R^2^ = 0.96 σ_0_ = 14.10	2.81 (0.32) R^2^ = 0.94 σ_0_ = 14.37	4.35 (0.25) R^2^ = 0.98 σ_0_ = 16.61	2.32 (0.26) R^2^ = 0.95 σ_0_ = 13.88	4.67 (0.39) R^2^ = 0.97 σ_0_ = 15.43
One month	3.59 (0.37) R^2^ = 0.95 σ_0_ = 13.13	2.65 (0.16) R^2^ = 0.98 σ_0_ = 12.26	2.46 (0.21) R^2^ = 0.97 σ_0_ = 15.63	4.06 (0.47) R^2^ = 0.94 σ_0_ = 17.25	5.03 (0.62) R^2^ = 0.93 σ_0_ = 18.54	2.89 (0.44) R^2^ = 0.90 σ_0_ = 15.41	2.77 (0.29) R^2^ = 0.95 σ_0_ = 18.94	4.82 (0.44) R^2^ = 0.96 σ_0_ = 16.76	3.24 (0.36) R^2^ = 0.94 σ_0_ = 19.76
Three months	2.61 (0.15) R^2^ = 0.98 σ_0_ = 13.89	2.95 (0.36) R^2^ = 0.93 σ_0_ = 14.89	3.75 (0.41) R^2^ = 0.95 σ_0_ = 12.79	5.66 (0.44) R^2^ = 0.97 σ_0_ = 17.13	9.41 (0.99) R^2^ = 0.95 σ_0_ = 20.26	3.23 (0.17) R^2^ = 0.99 σ_0_ = 17.28	2.46 (0.39) R^2^ = 0.89 σ_0_ = 19.01	3.30 (0.29) R^2^ = 0.97 σ_0_ = 14.05	2.27 (0.30) R^2^ = 0.76 σ_0_ = 17.02
Six months	2.93 (0.47) R^2^ = 0.89 σ_0_ = 13.83	3.15 (0.18) R^2^ = 0.99 σ_0_ = 11.79	2.88 (0.20) R^2^ = 0.98 σ_0_ = 10.87	2.63 (0.13) R^2^ = 0.99 σ_0_ = 13.79	5.50 (0.40) R^2^ = 0.98 σ_0_ = 17.16	3.95 (0.21) R^2^ = 0.99 σ_0_ = 16.41	2.35 (0.24) R^2^ = 0.95 σ_0_ = 11.46	3.35 (0.35) R^2^ = 0.95 σ_0_ = 14.15	2.79 (0.28) R^2^ = 0.95 σ_0_ = 16.42
One year	2.19 (0.16) R^2^ = 0.98 σ_0_ = 9.60	3.18 (0.38) R^2^ = 0.94 σ_0_ = 14.90	2.57 (0.28) R^2^ = 0.95 σ_0_ = 11.56	3.31 (0.58) R^2^ = 0.87 σ_0_ = 16.35	5.72 (0.60) R^2^ = 0.95 σ_0_ = 15.52	6.12 (0.86) R^2^ = 0.91 σ_0_ = 16.34	2.68 (0.17) R^2^ = 0.98 σ_0_ = 14.85	3.91 (0.35) R^2^ = 0.96 σ_0_ = 15.96	4.00 (0.25) R^2^ = 0.98 σ_0_ = 14.98

**Table 4 materials-13-05483-t004:** Fracture analysis of the test groups as a function of immersion duration. Prevalence of the defined fracture patterns is presented in percent (in total in brackets). Superscript letters identify fracture mechanism: a, adhesive; m, mixed; c, cohesive.

Fracture Analysis	EGCG 100 µM (wt/vol)		CHX 2% (vol/vol)		PA 3.75% (wt/vol)		HPN 5% (wt/vol)		Control (No Additive)	In Total
	Primer	Solution	Primer	Solution	Primer	Solution	Primer	Solution		
1 week	70% (14) a 30% (6) m 0% (0) c	55% (11) a 45% (9) m 0% (0) c	80% (16) a 20% (4) m 0% (0) c	45% (9) a 55% (11) m 0% (0) c	45% (9) a 55% (11) m 0% (0) c	60% (12) a 40% (8) m 0% (0) c	30% (6) ^a^ 60% (12) ^m^ 10% (2) ^c^	45% (9) ^a^ 55% (11) ^m^ 0% (0) ^c^	55% (11) ^a^ 45% (9) ^m^ 0% (0) ^c^	52.8% (95) ^a^ 46.1% (83) ^m^ 2.2% (2) ^c^
1 month	65% (13) a 35% (7) m 0% (0) c	60% (12) a 40% (8) m 0% (0) c	75% (15) a 25% (5) m 0% (0) c	30% (6) a 65% (13) m 5% (1) c	35% (7) a 65% (13) m 0% (0) c	60% (12) a 40% (8) m 0% (0) c	30% (6) ^a^ 65% (13) ^m^ 5% (1) ^c^	35% (7) ^a^ 55% (11) ^m^ 10% (2) ^c^	15% (3) ^a^ 85% (17) ^m^ 0% (0) ^c^	45% (81) ^a^ 52.8% (95) ^m^ 2.2% (4) ^c^
3 months	75% (15) a 25% (5) m 0% (0) c	85% (17) a 15% (3) m 0% (0) c	85% (17) a 15% (3) m 0% (0) c	60% (12) a 40% (8) m 0% (0) c	40% (8) a 60% (12) m 0% (0) c	60% (12) a 35% (7) m 5% (1) c	55% (11) ^a^ 35% (7) ^m^ 10% (2) ^c^	50% (10) ^a^ 45% (9) ^m^ 5% (1) ^c^	60% (12) ^a^ 40% (8) ^m^ 0% (0) ^c^	63.3% (114) ^a^ 34.4% (62) ^m^ 2.2% (4) ^c^
6 months	55% (11) a 45% (9) m 0% (0) c	70% (14) a 30% (6) m 0% (0) c	90% (18) a 10% (2) m 0% (0) c	60% (12) a 35% (7) m 5% (1) c	45% (9) a 50% (10) m 5% (1) c	70% (14) a 30% (6) m 0% (0) c	65% (13) ^a^ 35% (7) ^m^ 0% (0) ^c^	50% (10) ^a^ 50% (10) ^m^ 0% (0) ^c^	70% (14) ^a^ 30% (6) ^m^ 0% (0) ^c^	63.9% (115) ^a^ 35% (63) ^m^ 1.1% (2) ^c^
1 year	75% (15) a 25% (5) m 0% (0) c	50% (10) a 50% (10) m 0% (0) c	60% (12) a 35% (7) m 5% (1) c	45% (9) a 55% (11) m 0% (0) c	45% (9) a 55% (11) m 0% (0) c	45% (9) a 45% (9) m 10% (2) c	40% (8) ^a^ 55% (11) ^m^ 5% (1) ^c^	40% (8) ^a^ 60% (12) ^m^ 0% (0) ^c^	35% (7) ^a^ 65% (13) ^m^ 0% (0) ^c^	48.3% (87) ^a^ 49.4% (89) ^m^ 2.2% (4) ^c^
In total	68% (68) a 32% (32) m 0% (0) c	62% (62) a 38% (38) m 0% (0) c	78% (78) a 21% (21) m 1% (1) c	48% (48) a 50% (50) m 2% (2) c	42% (42) a 57% (57) m 1% (1) c	59% (59) a 38% (38) m 3% (3) c	44% (44) ^a^ 50% (50) ^m^ 6% (6) ^c^	44% (44) ^a^ 53% (53) ^m^ 3% (3) ^c^	47% (47) ^a^ 53% (53) ^m^ 0% (0) ^c^	54.7% (492) ^a^ 43.6% (392) ^m^ 1.8% (16) ^c^

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
