# Peer review of "Antioxidants and Collagen-Crosslinking: Benefit on Bond Strength and Clinical Applicability"

_materials, 2020, doi:10.3390/ma13235483_

Round 1

Reviewer 1 Report

The topic of the manuscrpt is interesting and considering the importance of the field to medicine around the world, there will be a large amount of potential readers.

However, this is a general "materials" journal, not a dedicated medical or dentistry focused one, with readers without a background in the techniques, etc. Therefore, the authors should attempt to open the background, materials and methods for any non-medical material scientist potentially interested in the paper. Take for instance the 1st sentences - "technique sensitive" here tells very little to non-dental medicine specialists. How does infiltration or solvent evaporation come to play?

Images of some examples of the adhesive and cohesive breaks would be nice, if not in manuscript, then in suplementary.

The data analysis and most of the dicsussion is presently only on the statistical basis, however, in a general material science journal, it would make sense to at least attempt to introduce some mechanistic discussion, the authors appear to provide some possible (potentially detectable?) bonding mechanisms that could be behind the differences, could these be assessed with some cleverly designed experiments? If it were possible to take the discussion from adhesive bonding to types of bonding (like detection of crosslinking, alterations of carbonly bonds..), it would make the presentation so much stronger.

Author Response

Cover Letter- Reviewer 1

All comments to the corresponding author have been addressed independently below. The author’s rebuttal is always in BLUE.

The author would firstly like to thank the reviewers’ for taking the time to read and critically appraise the manuscript and secondly to thank the reviewers’ for their positive constructive comments in improving the work.

Comments and Suggestions for Authors

The topic of the manuscrpt is interesting and considering the importance of the field to medicine around the world, there will be a large amount of potential readers.

However, this is a general "materials" journal, not a dedicated medical or dentistry focused one, with readers without a background in the techniques, etc. Therefore, the authors should attempt to open the background, materials and methods for any non-medical material scientist potentially interested in the paper. Take for instance the 1st sentences - "technique sensitive" here tells very little to non-dental medicine specialists. How does infiltration or solvent evaporation come to play?

Images of some examples of the adhesive and cohesive breaks would be nice, if not in manuscript, then in suplementary.

The data analysis and most of the dicsussion is presently only on the statistical basis, however, in a general material science journal, it would make sense to at least attempt to introduce some mechanistic discussion, the authors appear to provide some possible (potentially detectable?) bonding mechanisms that could be behind the differences, could these be assessed with some cleverly designed experiments? If it were possible to take the discussion from adhesive bonding to types of bonding (like detection of crosslinking, alterations of carbonly bonds..), it would make the presentation so much stronger.

Response to reviewer 1:

  • Page 3, line 49 to 70: Extensive restructuring of the introduction with background information about
  • Clinical significance and context of the study (extending lifespan of resin-based restorations)
  • Dentin morphology and structure and its (negative) influence on adhesion
  • Collagen structure
  • Extension of the graphics with fig1, Tab1 and Fig 2 to facilitate understanding of the methods for non-medical scientists
  • REM-pictures of the cohesive/adhesive fractures would have been an asset for further discussion, we’ll keep that in mind for following studies. In this case, REM analysis was not part of the study protocol, we analysed all specimen under a light microscope.
  • Page 17/18, line 483-494: introduction of fractographical analysis/ mechanistic perspective to the discussion

Thank you for your commitment and remarks,

Sincerely,

Franziska Beck, Nicoleta Ilie

Reviewer 2 Report

Dear authors,

Please, see my comments in the manuscript file.

The draft manuscript should be improved mainly in the introduction and material and methods section. It is not clear the methodology used as the material preparation. Dentin and collagen network should be deepening explain to understand the main contributors to this work. The material tested should be demonstrated with a figure/scheme as well as, characterization reinforced. The characterization of the materials should be improved too. The study is too theoretically and it is essential to be more illustrative. Which is the innovation of this study in the field? The authors should evidence of the role of this work in the dentistry-related area.

The ethics of the study was not written.

Best Regards

Author Response

Cover Letter- Reviewer 2

All comments to the corresponding author have been addressed independently below. The author’s rebuttal is always in BLUE.

The author would firstly like to thank the reviewers’ for taking the time to read and critically appraise the manuscript and secondly to thank the reviewers’ for their positive constructive comments in improving the work.

Comments and Suggestions for Authors:  The draft manuscript should be improved mainly in the introduction and material and methods section. It is not clear the methodology used as the material preparation. Dentin and collagen network should be deepening explain to understand the main contributors to this work. The material tested should be demonstrated with a figure/scheme as well as, characterization reinforced. The characterization of the materials should be improved too. The study is too theoretically and it is essential to be more illustrative. Which is the innovation of this study in the field? The authors should evidence of the role of this work in the dentistry-related area. The ethics of the study was not written. Additional changes marked in the manuscript.

Reviewer 2

  • Page 3, line 49 to 70: Extensive restructuring of the introduction with background information about
  • Clinical significance and context of the study (extending lifespan of resin-based restorations)
  • Dentin morphology and structure and its (negative) influence on adhesion
  • Collagen structure
  • Page 4, line 114-121: “The innovation of this study lays in the direct comparison of the antioxidant agents with each other and CHX. Further, it is analysed whether an application in an additional step might positively influence the substances effect on bond strength improving their efficiency compared to direct incorporation into the primer formulation. As far as we are informed, a direct comparison of both primer incorporation and pre-adhesive application of these agents in a clinically applicable bonding protocol, has not yet been conducted. Also, the long observation period without accelerated aging is innovative in that context.”
  • Page 4, line 123: corrected the spelling of null hypotheses; we used the plural of the word as there were two hypotheses stated
  • Page 5, line 130/131: more detailed description of the collection of the third-molars: all used teeth were sound (caries-free, no restorations of any kind), collected from specialized dental offices; detailed data about the donating patients is not applicable; though most teeth were probably extracted for orthodontical reasons in young patients
  • Generally: more detailed description of the substrate preparation and bonding process
  • Generally: subdivision of the materials and methods part with following subheadings dentin substrate preparation, bonding procedure, fractography, statistical analysis and ethical approval to make it easier for the reader to understand the methodology
  • Figure 1: illustration to clarify the experimental procedure with the teeth and the bonding process; for this purpose, a pictogram was chosen and created to make to study more illustrative for the reader
  • Table 1: characterization of the used adhesive and resin-composite
  • Page 6, line 161. Correction of number of groups per interval (incorrect in the first manuscript)
  • Page 7, line 167. “exposed area”; additional information about diameter and area size
  • Figure 2: Detailed description of the 9 test groups and the bonding sequence dependent on the test group
  • Page 10, line 241 to 246: ethics statement
  • Page 18, line 511-522: possible significance in dental clinical life; future perspectives for application

Thank you for your commitment and remarks,

Sincerely,

Franziska Beck, Nicoleta Ilie

Reviewer 3 Report

The manuscript assessed the effect on bond strength by the incorporation of Antioxidants on the dentine bonding process.

This research is under the scope of this journal; the topic is relevant for readers and this research deals with potentially significant knowledge to the field and an open a new way for future studies. Aim of this paper is quite interesting.

However, there are numerous issues in the present manuscript that need to be addressed before publication:

(Abstract) 

  • Must describe the Statistical analyze in this part, with the p-values!

(Introduction)

  • Identified the gap in literature!
  • Page 2 line 72 – correct “nullhypotheses”

(Statement of clinical Relevance)

  • What is the importance of this study for the clinical?
  • What this study has new?

(M&M)

  • Please include a statement in the Material and Methods section that the study has been approved by the institutional ethics committee and provide the number of the process.
  • How was the sample calculated? Did authors perform a power analysis to evaluate if this sample size was appropriate?
  • Did
  • When mentioning materials or devices: for some of them, you don't mention the manufacturer at all, for some you mention only the manufacturer, for some the manufacturer and country. But, the better way is mentioning the manufacturer and city/ country.
  • Perform a flowchart in a schematic of the experimental procedure with teeth. And add a table with the groups, materials, procedure sequence.

(Results)

- Improve the presentation of a narrative description of results. Also, the presence of so many commas makes reading very difficult.

- Page 5 line 195 - Correct “tabable 1”.

(Discussion)

  • The null hypothesis must be rejected in the discussion, and not in the conclusion.
  • Please, identified what was the limitation of this study? And also, future perspectives.

Author Response

Cover Letter- Reviewer 3

All comments to the corresponding author have been addressed independently below. The author’s rebuttal is always in BLUE.

The authors would firstly like to thank the reviewers’ for taking the time to read and critically appraise the manuscript and secondly to thank the reviewers’ for their positive constructive comments in improving the work.

Comments and Suggestions for Authors

Reviewer 3:

Abstract

  • Page 2, line 35: introducing the p-level for the study results, to emphasize the statistical significance; though, we refrained from backing up the reliability comparisons with Weibull’s m-moduli, due to the limited amount of words allowed in the abstract.
  •  

Introduction

  • Page 3, line 49 to 70: Literature gap: Extensive restructuring of the introduction with background information about
  • Clinical significance and context of the study (extending lifespan of resin-based restorations)
  • Dentin morphology and structure and its (negative) influence on adhesion
  • Collagen structure
  • Page 4, line 114-121: “The innovation of this study lays in the direct comparison of the antioxidant agents with each other and CHX. Further, it is analysed whether an application in an additional step might positively influence the substances effect on bond strength improving their efficiency compared to direct incorporation into the primer formulation. As far as we are informed, a direct comparison of both primer incorporation and pre-adhesive application of these agents in a clinically applicable bonding protocol, has not yet been conducted. Also, the long observation period without accelerated aging is innovative in that context.”
  • Page 4, line 123: corrected the spelling of null hypotheses; we used the plural of the word as there were two hypotheses stated

Materials and Methods

  • Page 9, line 217-221: Sample calculation + power analysis statement
  • Page 10, line 241 to 246: ethics statement
  • Generally: unification of the description of materials and devices as follows:

Name of material/device, manufacturer, city, country and if applicable LOT number

  • Generally: more detailed description of the substrate preparation and bonding process
  • Generally: subdivision of the materials and methods part with following subheadings dentin substrate preparation, bonding procedure, fractography, statistical analysis and ethical approval to make it easier for the reader to understand the methodology
  • Figure 1: illustration to clarify the experimental procedure with the teeth and the bonding process; for this purpose, a pictogram was chosen and created to make to study more illustrative for the reader
  • Table 1: characterization of the used adhesive and resin-composite

Results

  • Page 12: line 315-326: more narrative description of the Weibull results
  • Page 12, line 326: corrected spelling “table”

Discussion

  • Page 18, line 499- 510: The rejection of the null hypotheses now placed in the discussion not in the conclusion
  • Page 18, line 495- 498: Limitation of bond strength studies
  • Page 18, line 511- 523: stating possible significance for dental clinical life and suggesting future perspectives for application;

Thank you for your commitment and remarks,

Sincerely,

Franziska Beck, Nicoleta Ilie

Round 2

Reviewer 1 Report

The manuscript has been considerably improved by providing a clearer background for non-medical/stomatology specialists. Therefore, it has become suitable for publication in a general purpose materials journal.

Author Response

Thank you for your commitment and help.

Reviewer 2 Report

I recommend the publication

Author Response

Thank you for your commitment and help,

Reviewer 3 Report

This research is under the scope of this journal; the topic is relevant for readers and this research deals with potentially significant knowledge to the field and an open new way for future studies. Aim of this paper is quite interesting. 

The authors improved the quality of the manuscript.

Just need minor review on the references, 

 (References)

  • The references are not standardized - Please used the MDPI Style.
  • The names of journals were described in different ways (complete and others abbreviated)

Author Response

Reviewer 3:

The authors improved the quality of the manuscript.

Just need minor review on the references,

(References)

The references are not standardized - Please used the MDPI Style.

The names of journals were described in different ways (complete and others abbreviated)

Revisions:

  • We indeed used the MDPI Style. We re-checked all References, with special attention to journal names; abbreviated all journal names following ISO 4 standards; we changed some references manually, when EndNote output style did not result in correct format

Thank you for your commitment and help,